# Beneficial Exercises for Cancer-Related Fatigue among Women with Breast Cancer: A Systematic Review and Network Meta-Analysis

**DOI:** 10.3390/cancers15010151

**Published:** 2022-12-27

**Authors:** Yu-Chen Liu, Tsai-Tzu Hung, Sriyani Padmalatha Konara Mudiyanselage, Chi-Jane Wang, Mei-Feng Lin

**Affiliations:** 1School of Nursing, College of Medicine, National Taiwan University, Taipei 100, Taiwan; 2Nursing Department, National Cheng Kung University Hospital, Tainan 70101, Taiwan; 3Department of Nursing, College of Medicine, National Cheng Kung University, Tainan 70101, Taiwan

**Keywords:** breast cancer, exercise, cancer-related fatigue, palliative care inter-treatment, post-treatment, network meta-analysis

## Abstract

**Simple Summary:**

In this study, the aim of conducting the network meta-analysis (NMA) was to explore the efficacy of different exercises and their respective ranks in the context of reducing cancer-related fatigue (CRF) in women with breast cancer (BC) during the inter-treatment and post-treatment periods. We found that yoga, resistance exercise, and aerobic resistance were the top three recommended exercise modes for the purposes of reducing CRF during the inter-treatment period among women with BC; moreover, Qigong ranked last in terms of efficacy. Yoga, aerobic yoga, and aerobic resistance were the top three exercise options that should be performed in order to relieve CRF during the post-treatment period; conversely, relaxation ranked last in this respect. Aerobic plus resistance, resistance exercise, and yoga were conducted via using a supervisor, as well as short-time sessions. Objective measures are recommended in order to examine the causal relationship of vigorous exercise or yoga on physical functions. This was performed in order to obtain clarity in respect to the optimal therapeutic management strategies that will best aid with the reduction in CRF among women with BC.

**Abstract:**

Context: Cancer-related fatigue (CRF) is the most common cause of physical powerlessness in women with breast cancer (BC). The practice of continuous effective exercise is beneficial with respect to reducing CRF. Objective: To explore exercise efficacy and respective ranks with respect to reducing CRF in women with BC within the inter-treatment and post-treatment periods. Methods: Throughout this systematic review and network meta-analysis, articles published from 2000 to March 2022 were included. Article searches were conducted on the MEDLINE, Embase, CINAHL, and CENTRAL databases. Adult women with BC and different exercise programs were compared with those involved in regular care as a control during or after treatment. In addition, randomized controlled trials (RCT) were also included. A risk-of-bias assessment, with the revised Cochrane risk-of-bias tool regarding RCT and probability-based ranking, was established based on the surface under the cumulative rank (SUCRA) method via network meta-analysis. Results: a total of 5747 women with BC followed seven exercise interventions. Yoga (SMD = −0.54, 95% CI [−0.86, −0.22]) was ranked first (94.5%), with significant effects in respect of mitigating CRF, followed by resistance exercise (66.2%), and aerobic resistance (63.3%), while Qigong ranked last (22.2%) among the 36 RCT in inter-treatment. Aerobic resistance exercise (SMD = −0.52, 95% CI [−0.98, −0.07]) induced significant effects in respect of mitigating CRF in the 38 RCTs in the post-treatment period. Yoga, aerobic yoga, and aerobic resistance exercise ranked as the top three (75.5%, 75.0%, and 72.4%, respectively), whereas relaxation ranked last (38.8%) in the post-treatment period. Conclusion: Yoga, aerobic resistance, and aerobic yoga are recommended for the purposes of inter- and post-treatment exercise to reduce CRF in women with BC by enhancing their physical resilience and long-term quality of life.

## 1. Introduction

Cancer-related fatigue (CRF) is the most common symptom in women with breast cancer (BC) [1,2,3]. This type of fatigue also causes disease-related distress, individual perceptions of emotional/cognitive tiredness or exhaustion, as well as physical powerlessness [1,4,5,6,7]. The prevalence of CRF in women with BC can occur during treatment (inter) and after treatment (post). Previous studies have shown that almost half of women with BC have experienced fatigue symptoms during cancer treatments [8,9]. Further, 30% of female breast cancer survivors have endured fatigue after their cancer treatments [2,4,10]; in addition, their CRF is impacted by their daily physical activities (PA), such as exercise (group-based exercise programs, such as yoga and aerobic exercise) and participating in events. PA can play an important role in social connections and help in peer-influenced group experiences, as well as in the preservation of positive experiential qualities in external circumstances [11,12,13].

The influence of CRF can increase the burden on families and caregivers themselves. Furthermore, CRF may impact the possibility of early returns to work following cancer treatments due to the fact that it can continue more than five years after the completion of therapy [14,15,16,17,18]. Therefore, more attention is needed concerning this issue in order to help women with BC to better cope with CRF symptoms, both during and after treatments. In addition, proper fatigue prevention strategies should be adopted during treatments and after treatment periods.

Physiologically, CRF may be caused by cytokine-related transduction, which is achieved while also activating body immunocytes that are anticipating cancer cells [19,20,21,22]. Moreover, during structural changes, breast cancer treatments, such as surgery, chemotherapy, and radiotherapy, may cause physical damage to the body. Then, diverse changes related to pathological behaviors may thus begin, such as an increased metabolism, weakness, and/or exhaustion of the body’s functions. This could result in aggravated CRF and other negative symptoms [23,24]. However, certain selected exercise programs, such as yoga and resistance exercise, can lower cytokine-related transduction via physiological mediation and, thus, can result in a decrease in CRF [25,26,27]. Even so, clinicians should select the proper practice plan by considering the timing of different treatment periods and understanding the individual factors of different exercise programs [28]. Nonetheless, there is no ranking exercise recommendation for the purposes of reducing CRF in women with BC, whether in the context of during or after treatment.

A previous meta-analysis was conducted on this matter and, through this analysis, it was found that the effectiveness of unspecific exercise on CRF reduction during breast cancer treatment was 28%, thus slightly decreasing CRF [29]. In another study, during radiation therapy, it was found that exercise training had mild to moderate effects on fatigue reduction among women with breast cancer [30]. A previous RCT conducted using three exercise groups (25–30 min of aerobic exercise in three weekly sessions, 50–60 min of aerobic exercise, or a combined dose of 50–60 min of aerobic resistance exercise) during chemotherapy found no significant difference between the three groups [31]. However, Carayol (2013) included four types of exercise programs (yoga, aerobic exercise, resistance exercise, and aerobic resistance) in the inter-treatment period and found that metabolic-equivalent exercise and duration were positively correlated with the effects on CRF mitigation [32]. In addition, more extended moderate-intensity exercise significantly affects CRF mitigation [33].

Nevertheless, the impact of exercise dosage in respect of fatigue reduction in the post-treatment period remains unclear. Consideration of the treatment period and the characteristics of patients is crucial for the purposes of clinical practices when aiming to provide health education on exercise interventions in respect of relieving CRF from a long-term perspective. However, the type, intensity, and dosage fit for individuals in different treatment periods requires further clarification. It is also difficult to interpret the effectiveness of exercise programs from traditional pairwise meta-analyses that have inconsistent results. Therefore, studies in health-related fields have used network meta-analysis via direct and indirect comparisons of all available interventions in RCT studies [34,35]. This rigorous network meta-analysis (NMA) method enables multiple interventions to be jointly synthesized into a single model by providing a comprehensive ranking of CRF, as well as the effects of different exercise methods. In this manner, through the use of NMA, the aim is to explore the efficacy of different exercises and their respective ranks in the context of their ability in reducing cancer-related fatigue (CRF) in women with breast cancer (BC) during both inter-treatment and post-treatment periods.

## 2. Methods

This systematic review and network meta-analysis were registered with the International Prospective Register of Systematic Reviews (PROSPERO Reg No.–CRD42022333323) before data extraction to a priori written protocol existed. Further, the review and meta-analysis was conducted with the following PRISMA statement guidelines (see Appendix A) [36,37,38,39,40,41].

### 2.1. Search Strategy

Four databases (MEDLINE, Embase, CINAHL, and CENTRAL) were searched for studies published from 2000 to March 2022. Two controlled vocabularies from two databases, Embase and MEDLINE, were applied. The Emtree-controlled language included: ‘breast tumour, ‘Mastectomy’, ‘Exercise’, ‘kinesiotherapy’, ‘physical activity, ‘sport’, ‘training’, ‘dancing’, ‘muscle strength’, ‘Fatigue’, ‘Fatigue Severity Scale’, and ‘Fatigue Impact Scale’. Included MeSH terms were found in the following: ‘Breast Neoplasms’, ‘Oncology nursing’, ‘Mastectomy’, ‘Exercise’, ‘Exercise Therapy’, ‘Sports’, ‘Physical Exertion’, ‘Exercise Movement Techniques’, ‘Physical Fitness’, ‘Muscle Strength’, and ‘Fatigue’. Controlled vocabularies with English synonyms (i.e., pretext) were systematically applied as the search syntax in the four databases in order to avoid missing articles (see Appendix A). The grey literature was examined by using the governmental and social website; the literature examined ranged from studies from professional conference societies, conference books, and those obtained via the Google Scholar website. The cited published papers that were identified during the search process (as well as the reference lists of relevant articles), and previous systematic reviews were manually screened in order to confirm the sensitivity of the search strategy. Furthermore, we also supplemented the search results to the Endnote X20 bibliographical database.

### 2.2. Inclusion and Exclusion Criteria

The inclusion criteria were as follows: adult women with BC (P); the use of any exercise program as an intervention (I), and the results of these treatments were compared with those who received regular care as a control (C); and cancer-related fatigue (CRF) was assessed as a study outcome (O) by quantitative methods, and randomized controlled trials (RCT) were utilized. We have excluded exercise programs combining behavioral or cognitive-based approaches (such as mindfulness, meditation, improving sleep quality, or healing without touching the patients); different types of muscle intensity exercises were are conducted in outdoor spaces (such as lifting weights, working with resistance bands, heavy gardening—such as digging and shoveling, hill walking, and cycling). Additionally, review studies and articles in languages other than English were excluded.

### 2.3. Data Extraction and Article Selection

Two authors (Y.C.L. and T.T.H.) conducted data extraction independently, based on a codebook by reviewing three of the included studies. As per the codebook, the two authors extracted: (a) the study factors (i.e., authors, publication year, country, sample size with arms of the study, etc.); (b) the demographic data of populations (age, cancer stage, periods of therapy (pre- or post-treatment), etc.); (c) interventions, such as exercise types and the components of exercise (e.g., intensity, frequency, and duration); and (d) outcomes, such as fatigue measurement instruments and change mean scores in both pre- and post-intervention. In regard to the missing or incomplete data that were present in the included studies, the correspondent and/or first author were contacted based on the addresses provided in the published articles. When we had no response from the author, the mean was calculated from the median and CI in the available data of the included studies based on specific pre-existing studies [42]. This approach was adopted after a discussion was conducted among the other coauthors (S.P.K.M. and C.J.W.) in order to resolve variances. If there was any disagreement, a corresponding author (M.F.L.) was assigned to resolve the matter.

### 2.4. Classification in ‘Inter-Treatment’ or ‘Post-Treatment’

We followed NCCN guideline criteria in order to classify the treatment periods. However, we did not change original article descriptions regarding cancer treatment classifications while extracting the data [43]. The term “inter-treatment period” was defined as: all ongoing treatment, including periods of surgery, chemotherapy, and/or radiation therapy and hormone therapy, which typically finished within a year after the diagnosis of BC. On some occasions, hormone therapy could continue post-treatment. However, if it is in within the first year and was combined with other treatments, it was then considered an inter-treatment. Additionally, a treatment in the presence of metastasis or cancer recurrence was classified as an ‘inter-treatment period’. ‘Post-treatment period’ was defined as: the survival time (survivorship) following the previously mentioned treatments’ completion; however, the practice of hormone therapy as a regular maintenance dose being continued following the treatment completion was seen as an exception. Furthermore, studies with both types of patients were classified according to most of the patients. Studies among the patients receiving androgen suppression therapy without chemotherapy or radiotherapy were placed into the category of ‘post-treatment period’.

### 2.5. Exercise Classification

We constructed certain operational definitions in order to classify exercise modes while using ACSM exercise prescription guidelines and their defined level of intensity. There were seven exercise modes (resistance exercise, aerobic resistance, aerobic, aerobic yoga, yoga, Qigong (Tai Chi), and relaxation) that were utilized in the control group [44,45]. The resistance exercises included walking, running, stair climbing, and swimming indoors. Additionally, workouts were required to be hard enough to entail sweating. This was achieved by involving more muscles with external resistance and for a duration of 30 min per session at five days per week, or 20 min of more vigorous activity at three days per week [46,47,48]. Aerobic resistance exercise is understood to be any movement that uses large muscle groups. It can be continuous and rhythmic, such as dancing, short-distance jogging, swimming, and walking/brisk walking [49,50]. Aerobic yoga was required to stimulate the chest wall (including the heart and lungs) in order to predict the maximum heart rate for at least six weeks, three days a week, for 30 min each day at the exercise level of 60% to 70% [51,52]. Conversely, in the context of aerobics, yoga, Qigong (Tai Chi), and relaxation exercises, we combined psychological strategies with definitions according to the selected articles that match with the original definitions [53,54,55].

### 2.6. Risk of Bias Assessment

Two reviewers (Y.C.L. and T.T.H.) evaluated the risk of bias (RoB 2.0), independently, according to the revised Cochrane risk of bias tool for the purposes of randomized trials [56,57]. The specific areas that were assessed for quality and bias appraisal were: (a) allocation, (b) performance, (c) follow-up, (d) measurement, (e) reporting, and (h) overall risk of bias. Each area was rated as at either a “low,” “unclear”, or “high risk” of bias. Disagreements were judged and solved by the third reviewer. Statistically, publication bias was determined via the use of funnel plots. In addition, the Q statistics obtained from the Egger method were used to determine the comparisons of the active interventions against the control groups. Moreover, the results for cancer-related fatigue regressed the effect estimates on their standard errors. Further, they were weighted by the inverse of the variance via the software of Stata 16 and a visual examination of the funnel plot [57,58,59].

### 2.7. Statistical Analysis

The NMA was performed under conditions where a minimum of two studies possessed homogeneity in terms of population, intervention, outcome, or any other related factors [57,60]. As the various instruments used within the included studies measured fatigue, a standardized mean difference (SMD) with a 95% confidence interval (CI), as well as random effect models, were adopted in order to compare the relative effectiveness of the different interventions that are under examination. The design-by-treatment interaction model was used to examine the inconsistency of the NMA being conducted [61]. Generalized linear mixed models were adopted in order to assess the inconsistent interactions between the study design and exercise types. When *p* < 0.05, this indicated a significant inconsistency between the study designs and exercise types [62,63]. The design-by-treatment inconsistency of this NMA was nonsignificant (*p* = 0.99). Thus, we assumed the included studies were consistent. Heterogeneity was evaluated using Higgins I^2^ for each pairwise comparison [64]. A probability-based ranking was established based on the surface under the cumulative ranking (SUCRA) method, as well as on the probability best and mean rank [65,66] approaches. The SUCRA of a specific intervention was construed as the average proportion of interventions vs. the intervention under consideration. Ranking probabilities were assessed by each exercise type per each possible ranking that inclusively ranged from 0% to 100%. Sensitivity analysis was conducted, excluding studies with less than 25 patients per intervention arm [67]. The publication bias of included studies [68] and network metanalysis data were analyzed using RevMan 5.3 and Stata 16 software.

## 3. Results

### 3.1. Study Selection and Characteristics

A total of 3308 articles were identified from four databases and from other additional sources. After removing 1168 duplicates, a total of 184 full-text articles were assessed for eligibility; in addition to this, 74 RCTs were also included in this study (see Appendix A).

The 74 included RCTs are described in Table 1. Studies were conducted in 23 countries. Thirty studies were conducted in the United States (USA); six studies were conducted in Germany; four studies were conducted in Korea and Turkey; three studies were conducted in Taiwan, Brazil, and the Netherlands; two studies were conducted in Canada, China, Sweden, United Kingdom (UK), and Australia; and one study was conducted in Denmark, India, Iran, Latvia, Norway, Scotland, Finland, Japan, Malaysia, Spain, and Thailand, respectively. Studies were published across the period from 2000 to 2022.

A total of 5820 women with breast cancer underwent seven exercise interventions. A total of twenty-one studies were conducted on resistance exercise, 26 studies on aerobic resistance, 20 studies on aerobic, one study on aerobic yoga, 17 studies on yoga, three studies on Qigong (Tai Chi), and six studies on relaxation. It must also be noted that sixty-six control groups in the inter-treatment and post-treatment periods (Table 1) were utilized in this study.

Thirty-six studies were conducted on inter-treatment periods. We found that 22 studies were conducted during chemotherapy, seven were conducted during radiation therapy, and seven studies did not specify treatment types. Thirty-one included studies with participants in the early stages of breast cancer, two included all cancer stages, and three did not provide information regarding the cancer stages in respect of their participants in the inter-treatment periods (Table 1).

Thirty-eight studies were conducted during the post-treatment periods. Thirty studies included early stage breast cancer, two included all cancer stages, one included stage II–IV, and five did not provide information on the cancer stage of the participants in post-treatment periods (Table 1).

The network geometries show the interactions in these trials based on the effects of the exercise programs in respect of the CRF in women with breast cancer. The size of the nodes represents the proportional number of participants. A larger node indicated a more significant number of participants. The number of studies directly compared to its effects is represented by the thickness of the lines connecting two nodes. Multiple comparisons were performed after examining the existence of closed loops. Seven and eight intervention types formed seven closed loops during the inter- and post-treatment periods, respectively (Figure 1).

### 3.2. Rob of Included Trials

Overall, 22 studies (29.7%) possessed a high risk of bias, 41 studies (55.4%) had some concerns, and only nine studies (12.2%) possessed a low risk of bias. Around 18.9% (14 studies) of the included RCTs had a high risk of reporting bias, which was mainly formed by the use of inappropriate statistical methods on a small sample size. A missing outcome data bias was shown in eight of the included studies (10.8%). Due to the nature of exercise programs, participants may have failed to complete the whole program due to their physical condition and/or privacy schedule (see Appendix A). The symmetric funnel plots of both inter- and post-treatment periods indicate low publication biases. No small-study effects were found by the use of Egger’s test-based funnel plots (*p* =0.811 and *p* =0.740 for inter- and post-treatment, relatively) (Figure 2).

### 3.3. Exercise Efficacy and Ranking during Inter-Treatment Period

A total of 36 RCTs were included in 16 countries. Twelve studies occurred in the US, four studies occurred in Germany; three studies occurred in Taiwan; two studies each occurred in Canada, China, Netherlands, Sweden, and the UK; and one study occurred each in Denmark, India, Iran, Korea, Latvia, Norway, and Scotland, respectively. These studies were analyzed in terms of exercise efficacy in respect of mitigating CRF in women with BC during inter-treatment periods. A total of 3074 women with BC participated in six exercise programs. Among the six exercises there were: 11 studies were conducted on resistance exercises with 540 participants; 10 studies were conducted on aerobic resistance with 367 participants; 10 studies were conducted on aerobic exercise with 375 participants; 10 studies were conducted on yoga with 437 participants; one study was conducted on Qigong with 49 participants; and three studies were conducted on relaxation with 170 participants. In addition, 32 studies included a control group with 1136 participants labeled as treat-as-usual, waiting list, or education groups (see Table 1).

Yoga was mainly conducted under supervision (87.5%), with 90 min/session (ranging from 60 to 120 min/session) once a week (ranging from one to five sessions/week) and lasting for eight to 12 weeks (ranging from 8–26 weeks). The aerobic resistance exercise was conducted under supervision, with 30–50 min/per session (ranging from 20 to 80 min/per session), thrice a week (ranging from one to five sessions/per week), and lasting for 12 weeks (ranging from 4–36 weeks). Only one study was conducted in regard to aerobic yoga (see Table 1).

The interval plot of intervention effect size, mean, and 95% CIs is shown in Figure 3. When compared with routine care, yoga (SMD = −0.54, 95% CI [−0.86, −0.22]) possessed significant effects in respect of CRF. However, others did not find significant differences when conducting pairwise comparisons. In addition, no significant difference was found between the exercises and the control group. The efficacy cumulative rank probabilities among different interventions are demonstrated in Figure 4. The results showed that yoga was deemed to be the first-rank, most effective exercise (94.5%), followed by resistance exercise (66.2%), and aerobic resistance (63.3%). Qigong ranked last (22.2%). The contribution plot of the network suggests that the comparison of Qigong and the control group possessed the most significant contribution in the entire network (14.0%), followed by the comparison of resistance exercise and the control group (12.5%). The contribution of the other comparisons ranged from 4.1% to 9.8% (Figure 5).

### 3.4. Exercise Efficacy and Ranking in Post-Treatment

A total of 38 RCTs in 12 countries (18 in the USA; four in Turkey; three each in Brazil and Korea; two each in Australia and Germany; and one each in Finland, Japan, Malaysia, Netherlands, Spain, and Thailand) have analyzed the exercise efficacy in CRF in women with BC during post-treatment periods. A total of 2673 women with BC followed seven exercise types. A total of 11 studies were conducted with respect to resistance exercises with 209 participants, 14 studies were conducted on aerobic resistance with 391 participants; 10 studies were conducted on aerobic exercise with 574 participants; one study was conducted on aerobic yoga with 19 participants; eight studies were conducted on yoga with 206 participants; two studies were conducted on Qigong with 47 participants; and three studies were conducted on relaxation with 62 participants included in post-treatment periods. Thirty-one studies included control groups labeled as treat-as-usual and psychoeducation with 1165 participants (see Table 1).

Yoga was conducted mainly by the use of a supervisor (88.9%), with 60 min/session (ranging from 50 to 90 min/session), once a week (ranging from three to four sessions/week), and lasting for 12 weeks (ranging from 6–12 weeks). The aerobic resistance was also under supervision (80.0%), with 50–60 min/session (ranged from 10 to 60 min/session), twice to thrice a week (ranging from one to 12 sessions/week), and lasting for 12 weeks (ranged from 5–27 weeks). The resistance exercise used a supervisor (81.8%), with 30 or 60 min/per session (ranging from 20 to 60 min/per session), twice a week (ranging from one to five sessions/per week), and lasting for 12 weeks (ranged from 6–18 weeks) (see Table 1).

Figure 3 presents the interval plot for an intervention effect size with SMD and 95% CI compared with routine care. Only aerobic resistance exercise (SMD = −0.52, 95% CI [−0.98, −0.07]) significantly affected CRF. No significant differences were found in other comparisons. The SUCRA probabilities among different interventions are demonstrated in Figure 4. The results mostly showed that yoga, aerobic yoga, and aerobic resistance exercises appeared to be ranked as the top three (75.5%, 75.0%, and 72.4%), respectively. Relaxation ranked last (22.9%). The contribution plot of the network suggests that the comparison of aerobic exercise and the control group possessed the most significant contribution in the entire network (14.2%), followed by the comparison of resistance exercise and aerobic yoga (11.7%). The contribution of the other comparisons ranged from 2.2% to 9.8% (Figure 5).

## 4. Discussion

CRF is the most common symptom in women with BC during the entire lifecycle of the disease. This is the first network meta-analysis to explore exercise efficacy and comprehensively rank women with BC based on their treatment periods (inter and post). Seventy-four studies were conducted across twenty-three countries worldwide from 2000 to 2022. Seven exercise modes were analyzed: resistance exercise, aerobic resistance, aerobic exercise, aerobic yoga, yoga, Qigong (Tai Chi), and relaxation.

The most effective high-rank exercise interventions were more beneficial to prescribe exercise in order to reduce CRF, such as oncologists, nurse practitioners, physiotherapists, family practitioners, clinicians from palliative care, and women with BC. Additionally, the patient’s preferences, contraindications, the availability of services, and the costs of the interventions are beneficial with respect to the patients’ participatory shared decision-making process. Therefore, the findings in this study help to fulfil the gap between patients and healthcare practitioners’ priorities in respect to evidence-based practice strategies regarding exercise interventions in respect to mitigating CRF during and after BC treatments. Currently, a variety of exercise programs have been provided by healthcare professionals as a trending effect in respect of reducing CRF. Meanwhile, different interventions showed moderate-to-high effects. By using high-rank exercise it will be easier for patients to select a modality that is most convenient for themselves as individuals. For example, if women with BC do not want to participate in resistance exercise in order to reduce CRF, then they may choose aerobic yoga or yoga as an alternative based on the findings of this study.

### 4.1. Exercise Efficacy and Ranking in Respect of the Inter-Treatment Period

Thirty-six studies utilized six types of exercise modes. We found that yoga was the first-rank exercise in respect of CRF. Then, resistance exercise, followed by aerobic resistance according to the SUCRA values, were derived from the indirect comparison that was conducted during inter-treatment periods. These NMA findings are consistent with previous traditional meta-analyses that indicated that yoga exercise positively reduces CRF in women with BC when compared with the application of standard care or attentional control during treatment. However, those studies’ quality did not meet the advanced status of acceptable quality [69,70,71]. However, certain meta-analyses found nonsignificant effects in respect to yoga reducing CRF in women with BC during treatment due to proportional differences in the publication year [72]. This NMA included higher proportions of studies published after 2009 (92%). Additionally, we conducted both direct and indirect pairwise comparisons, which yielded more strengths in respect to providing evidence than those found in more traditional meta-analyses.

Furthermore, most of the included individual RCTs recommended that yoga exercise is the most suitable intervention in respect to CRF due to its benefits in aiding the increased physical functions of muscles, tendons, and ligaments—which are of particular benefit to patients with BC, as they achieve a steeper degree of cortisol by the end of the exercise [73,74,75,76]. Furthermore, certain inconsistent findings were reported in previous meta-analyses stating that supervised aerobic resistance exercise, primarily, can moderately relieve CRF in women with BC [77]. Having said that, when conducting this NMA, it was found that aerobic resistance exercise was the second most effective exercise with respect to mitigating CRF when compared with the control group during inter-treatment [78]. Resistance exercise was found to be the third-most beneficial in a row in our NMA. This is due to the fact that the physical fitness deformities of women with BC during treatment were a mitigating factor as to whether resistance training was appropriate to conduct for patients with physical limitations due to the disease/surgery/treatment. Furthermore, it may cause lymphoedema, fitting-oedema, or certain physical complications, such as cytokine-related transduction while activating body immunocytes that anticipate cancer cells during aggressive exercises in a short time. Moreover, additional discomfort for cancer patients is also a possibility [19,79,80].

A recent NMA study analyzed the effect of exercise and other non-pharmaceutical interventions with respect to CRF in patients during and after cancer treatment. It was mentioned in the study that aerobic resistance exercise was first-rank and yoga was second-rank. This was determined by combining all cancer patients during treatment periods [81]. However, their findings are inconsistent with ours due to the population of women with BC. Therefore, these NMA findings can only apply to women with BC, which is the most common female cancer worldwide [82,83]. Our findings indicated that Qigong (Tai Chi) was less effective in reducing CRF during treatment periods. This finding, however, was inconsistent with the findings in previous meta-analysis studies that were conducted on lung and breast cancer during chemotherapy [84,85,86]. However, they did not perform a pairwise comparison of each exercise, as is the case in this study. Furthermore, there were no NMA studies that were conducted on the effectiveness of Qigong (Tai Chi), specifically with respect to mitigating CRF. Therefore, future studies should focus on the effectiveness of Qigong (Tai Chi) with respect to mitigating CRF in women with BC in order to more robustly support our, and other, study findings. Moreover, there are less performances of Qigong (Tai Chi) during treatment due to existence of other high vigorous exercises, such as resistance exercise, which causes increased function of cytokine-related transduction [87,88]. Per our findings, resistance exercise, aerobic resistance exercise, and yoga are all conducted under supervision and 60 min/per session for one to 12 sessions. However, during treatment, resistance exercise was conducted for 30 or 60 min/per session twice a week. These findings are similar to ACSM’s guidelines [53,89] for normal healthy adults’ exercise prescription, which still needs to be precise for BC populations. Hence, these findings may consider a prescribing protocol for BC.

### 4.2. Exercise Efficacy and Ranking in the Post-Treatment Period

Seven exercise modes (resistance exercises, aerobic resistance, aerobic exercise, yoga, aerobic yoga, Qigong, and relaxation) have been used in 38 studies during post-treatment periods. Yoga is first-ranked. Aerobic yoga and aerobic resistance were ranked second and third in the row. A recent NMA found similar ranks conducted using non-pharmaceutical interventions for patients with all cancers during post-treatment [81]. Some traditional meta-analyses also mention consistent results [77,90]. However, they did not specify BC. BC is a global life-threatening disease that significantly influences women’s health in a perceptively long-term manner [91,92]. Therefore, our study is distinguished from preferring different kinds of exercise for women with BC by completing long-term health effects on women’s health. Additionally, those high-rank exercises can be prescribed by health care professionals, precise case managers, nurse practitioners, physiotherapists, and other respective care specialties, such as palliative care professionals, for their personalized exercise care plans in women with BC.

As per the previous literature, performing aerobic resistance exercise entails better effects with respect to improving the function of the skeletons and muscles, maintaining muscle mass, and balancing the degree of cytokine [93,94,95,96]. This is possible due to the exercise type’s wide-ranging intensity. This exercise should be tailored and supervised by professional providers during the inter-treatment period for the purposes of safety considerations. However, this NMA showed that it could be beneficial for both inter- and post-treatment when paired with a high-ranking exercise mode; therefore, aerobic resistance exercise can be recommended for women with BC in their inter- and post-treatment periods. Yoga exercise and aerobic yoga play significant roles with respect to the reduction in CRF in women with BC during post-treatment. As such, it needs to be considered in advance for future exercise guidelines due to the demand and availability of the abovementioned exercises around the world, as well as new trending exercises [97,98]. Duration, frequency, supervised base exercise, and intensity have all been declined intervention profits in women with BC [99]. However, future studies are required to further analyze the effectiveness of specific exercise interventions. This is required due to the fact that the effect of prescribed exercise adherence, with respect to CRF in women with BC, remains unknown.

Through conducting this NMA, it was found that the robustness of the evidence of the ACSM guidelines [89] recommend that healthy adults participate in a moderate-intensity aerobic or vigorous-intensity exercise in order to reduce pain during medical procedures. Thus, our findings with respect to the BC population also parallel the results found during post-treatment—such as yoga, aerobic resistance exercise, and resistance exercises—in that they were mainly conducted by the supervisor. Frequently, resistance exercise was performed for 30 or 60 min/session (ranging from 20 to 60 min/session). In addition, it would be recommended twice a week to follow for future exercise prescriptions in women with BC survivors as a long-term perspective.

Another essential finding in our NMA was that relaxation exercises were low in number during inter- and post-treatment due to the fact that certain relaxation sessions were added to the personalized vigorous exercise plan and did not consider a unique exercise method. However, this is inconsistent with previous NMA studies with respect to exercise effects on nonpharmacological intervention in cancer patients [81]. However, devoted relaxation time or sessions should be kept, or more time should be arranged for the performance of high-rank exercises, such as aerobic resistance, yoga, or aerobic yoga in order to the enhance an intervention’s effectiveness with respect to mitigating CRF in women with BC [100,101].

### 4.3. Study Strengths and Limitations

This study possesses a few strengths. Firstly, this is the first study to explore exercise efficacy and rank in respect of mitigating CRF in women with BC using a NMA method. We included 74 recently published RCTs within 23 countries around the world. Therefore, the comprehensive efficacy of exercise and ranking can be used as reference material in order to develop and update exercise protocols. Secondly, statistical models of the design-by-treatment interaction model were of a high standard in order to examine the inconsistency and consistency findings through the application of a NMA. The results from the SUCRA method, probability test, and mean rank were all more accurate and reliable for the purposes of clinical practice. Therefore, this study’s findings can be used as a preferred reference and prescribed protocol for the purposes of exercise interventions with respect to mitigating CRF in women with BC. However, we recommend conducting physiological mechanisms of low- and high-vigorous exercise with respect to mitigating CRF in women with BC, such as resistance exercise to relaxation. Finally, this study included seven methods recategorized as resistance exercise, aerobic resistance exercise, aerobic yoga, yoga, Qigong (Tai Chi), and relaxation as based on the level of intensity as per the ACSM exercise prescription guidelines [44,89]. Therefore, these findings are of utmost importance for the purposes of developing exercise protocols and strategies in order to modify the perception of exercise with respect to mitigating CRF in women with BC. Furthermore, we encourage continuing exercise programs, such as aerobic yoga, yoga, or aerobic resistance exercise, in order to increase the quality of life through women’s health as a palliative care improvement in the community.

There are certain limitations to this study. First, our included study did not measure biomedical parameters, such as serotonin levels or cytokine levels, after exercise intervention, in order to view the changes in biological parameters via physical functions. Therefore, it is recommended to consider biological parameters in order to examine the causal relationship between yoga or vigorous exercise on physical functions. Second, we notice that 22.9% of studies showed a high RoB due to the large number of studies containing a small sample, which thus made it hard to also reach the blinding strategy. Third, studies have reported high variable retention rates. Therefore, we conducted pairwise comparisons, as well as observed moderate-to-high clinical and statistical heterogeneity. Then, we conducted a sensitivity analysis. This was performed, such that it will not significantly influence our study findings. However, we recommend conducting long-term follow-up studies every five or ten years in order to strengthen the empirical evidence regarding the effect of exercise in respect of mitigating CRF in women with BC by implementing a novel NMA study.

### 4.4. Clinical Implications

Our study findings would be supported by the clinical practice guideline for CRF in women with BC. This is because we have to rank the top three most effective exercise modes that can be used in inter- and post-treatment periods. There is also variance in depending on the BC treatment status of the patients. During inter treatments, yoga possesses the most significant benefits with respect to mitigating CRF, followed by aerobic resistance, and resistance exercise under the watch of a supervisor. Therefore, front-line clinical practice and rehabilitation or palliative care professionals should consider these high-rank exercise interventions for the purposes of further application in order to enhance their physical resilience and QoL during the BC treatments.

Post-treatment periods are essential for women with BC due to entering in their daily routing works early. Therefore, the most vigorous and rhythmic exercises, such as aerobic yoga, yoga, or aerobic resistance, were more beneficial instead of relaxation or Qigong (Tai Chi). In addition, short periods with supervised-based practice are recommended to reduce CRF in women with BC. Therefore, the palliative healthcare professionals in the community should consider and prescribe the highest-ranking exercise intervention treatment. However, some of the exercise modes included in this NMA exhibited similar effects and SUCRA values. Therefore, health professionals should make a shared decision between coworkers, patients, and families in order to choose between different exercise modes—such as either yoga or aerobic yoga, etc. [87,88,89,90,91,92,93,94,95,96,97,98,99,100,101,102,103,104].

Furthermore, our study findings can be referenced by the patient themselves so as to choose the list of effective exercise alternatives to reduce CRF in women with BC. Therefore, it will raise adherence to the exercise intervention. However, there are some negative influences of CRF on avoiding regular exercise in women with BC and survivors [81,105,106]. Therefore, it is essential to understand individual coping status and tolerance of CRF, as well as re-evaluating regular follow-up processes by the healthcare team.

### 4.5. Implications for Research

As per a comprehensive evidence search in clinical practice, we would suggest the most effective exercise based on the available studies. Nevertheless, we still require a straightforward exercise prescription model with a causal effect with respect to reducing CRF in women with BC. Such effects may range from sputum or blood for serotonin levels, or they may involve cytokine levels after exercise intervention, which are limited in order to see the actual reason for the exercise reducing CRF in women with BC. Additionally, the effect of exercise on the inflammation–immunity axis is still a complex phenomenon and still lacks objective and subjective evidence [87,107]. Persons’ perception of high-intensity to moderately vigorous exercise, as well as the adaptation to exercise, need to be evaluated [108,109]. Seeing the individual success of reducing CRF in BC women through some specific exercise programs, such as supervised or group base exercise programs, will be challenging to conduct in the future due to special epidemic conditions, such as COVID-19 [110]. Therefore, the effectiveness of distance-based exercise programs, such as online or game application exercises, must be accounted for in future studies. Additionally, it is necessary to conduct exercise adherence and economic evaluation of exercise, such as cost-effectiveness, cost-utility, or cost-benefit analysis studies [87]. Finally, we recommend psychrometries analysis for the purposes of questionnaire validation in order to measure CRF in respect of BC.

## 5. Conclusions

Through conducting this NMA, it was found that yoga, aerobics resistance, and resistance exercises were the top three most effective exercises during the inter-treatment period. Furthermore, yoga, aerobic yoga, and aerobic resistance exercise were the top three (75.5%, 74.8%, and 72.2%, respectively) during the post-treatment period. Most vigorous exercise modes, such as aerobic resistance, resistance exercise, and aerobic yoga, were conducted in conjunction with a supervisor. Resistance exercise was conducted in short-time sessions. Based on our evidence, healthcare professionals can prescribe and follow the most suitable exercise mode in order to relieve CRF during inter-treatment or post-treatment by the improving of quality of life and physical resilience among women with BC.

## Figures and Tables

**Figure 1 cancers-15-00151-f001:**
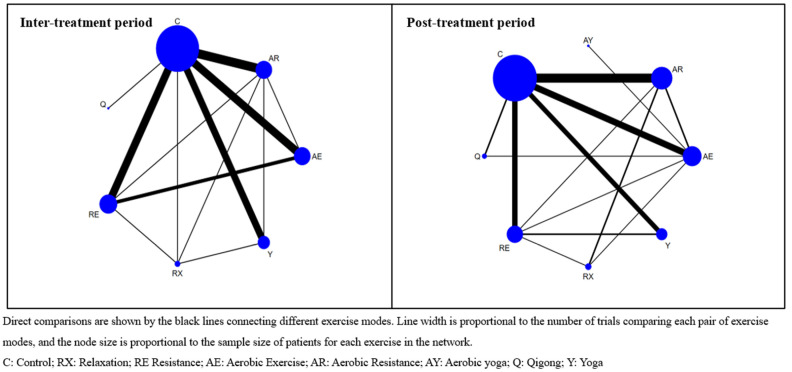
Network map of the included studies with the number of participants during inter- and post-treatment exercise modes.

**Figure 2 cancers-15-00151-f002:**
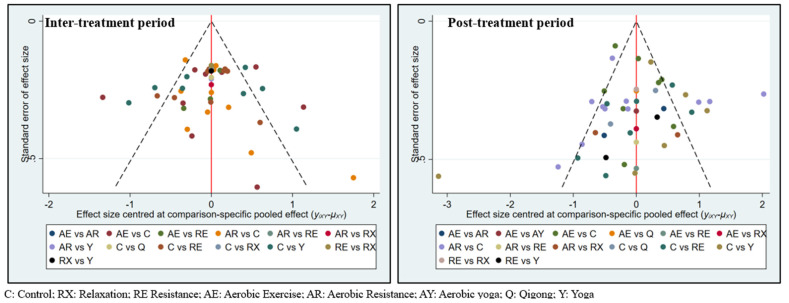
A funnel plot of the studies’ reporting publication bias according to the application of Egger’s test, which was based on exercise mode used during inter- and post-treatment periods.

**Figure 3 cancers-15-00151-f003:**
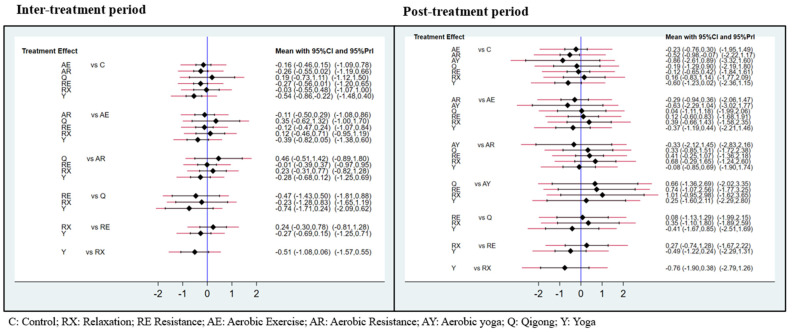
The interval plot of exercise effect size, mean, and 95% CI during inter- and post-treatment periods.

**Figure 4 cancers-15-00151-f004:**
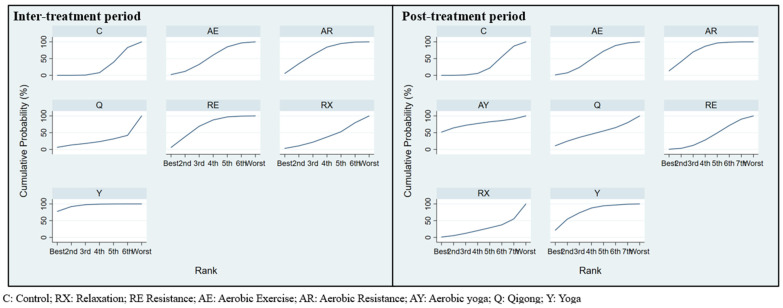
The efficacy cumulative rank probabilities among different exercise modes during inter- and post-treatments periods, whereby the probability for the exercise’s modes is indicated.

**Figure 5 cancers-15-00151-f005:**
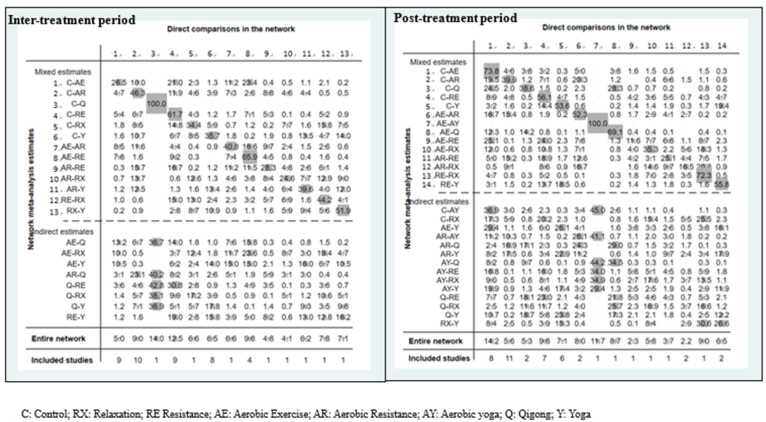
The contribution plot of the network suggestion for indicating the probability in respect of the best exercise mode, in descending order.

**Table 1 cancers-15-00151-t001:** Summary of characteristics of the included studies in regard to inter-treatment and post-treatment.

Author Year	Treatment Phase	Country	Stage	Age Mean ± SD	CRF Measurement	Sample Size & Exercise Interventions	Findings Mean (SD) of CRF Score Effect Size (95% CI)
Al-Majid 2015	Inter-treatment (CT)	USA	I–II	AE: 47.9 ± 10.4 C: 52.7 ± 10.7	Revised Piper Fatigue Scale (PFS) (SD is calculated by SE)	**AE (n = 6)** Length: 20–40 min/session Frequency: two to three sessions/week Duration: 12 weeks Supervised: no **Control (n = 6)** Usual care	**AE:** 3 (1.96) **C:** 4.6 (2.2) **Hedges’ g:** −0.71 (−1.89, 0.47)
Battaglini 2006	Inter-treatment (OP, CT, or RT)	USA	NI	AE+RE: 57.5 ± 23 C: 56.6 ± 16	PFS	**AE+RE (n = 10)** Length: 60 min/session Frequency: two sessions/week Duration: 15 weeks Supervised: yes **Control (n = 10)** Usual care	**AE+RE:** 0.84 (1.13) **C:** 3.23 (1.16) **Hedges’ g:** −2.00 (−3.12, −0.88)
Bolam 2019	Inter-treatment (CT)	Sweden	I–IIIA	RE: 52.7 ± 10.3 AE: 60 ± 10.3 C: 57 ± 10.2	PFS	**RE (n = 65)** Length: 60 min/session Frequency: two session/week Duration: 16 weeks Supervised: yes **AE (n = 60)** Length: 20 min/session Frequency: two session/week Duration: 16 months Supervised: yes **Control (n = 57)** Usual care	**RE: 3.12** (3.03) **AE**: 3.18 (2.77) **C:** 3.98 (3.05) **Hedges’ g:** **RE:C** −0.28 (−0.64, 0.07) **AE:C** −0.27 (−0.64, 0.09) **RE:AE** −0.02 (−0.37, 0.33)
Campbell 2005	Inter-treatment (OP, CT, or RT)	UK	Early stage	AE+RE: 48 ± 10 C: 47 ± 5	PFS (Using changed score)	**AE+RE (n = 10)** Length: 10–20 min/session Frequency: two sessions/week Duration: 12 weeks Supervised: yes **Control (n = 9)** Usual care	**AE+RE:** −2.11 (2.3) **C:** −0.25 (2.5) **Hedges’ g:** −0.74 (−1.68, 0.20)
Cešeiko 2019	Inter-treatment (OP, CT, or RT)	Latvia	I–III	RE: 48.2 ± 6.7 C: 49.0 ± 8.0	European Organization for the Research and Treatment of Cancer—Quality of Life (EORTC QoL C30)	**RE (n = 27)** Length: 20 min/session Frequency: two session/week Duration: 12 weeks Supervised: yes **Control (n = 28)** Usual care	**RE:** 25.5 (15.5) **C:** 36.8 (16.7) **Hedges’ g:** −0.69 (−1.24, −0.15)
Chandwani 2010	Inter-treatment (RT)	USA	0–III	Y: 51.4 ± 8.0 C: 4.0 ± 10.0	Brief Fatigue Inventory (BFI) (SD is calculated by SE)	**Yoga (n = 27)** Length: 60 min/session Frequency: two sessions/week Duration: six weeks Supervised: yes **Control (n = 31)** Waiting list	**Y:** 1.9 (3.64) **C:** 2.5 (4.45) **Hedges’ g:** −0.14 (−0.66, 0.37)
Chandwani 2014	Inter-treatment (RT)	USA	0–III	Y: 52.4 ± 9.8 C: 52.1 ± 9.8	BFI	**Yoga (n = 53)** Length: 60 min/session Frequency: three sessions/week Duration: six weeks Supervised: yes **Control (n = 54)** Waiting list	**Y:** 2.9 (0.3) **C:** 3.2 (0.4) **Hedges’ g:** −0.84 (−1.24, −0.45)
Chaoul 2018	Inter-treatment (CT)	USA	I–III	Y: 49.5 ± 9.8 RX: 50.4 ± 10.3 C: 49.0 ± 10.1	BFI	**Yoga (n = 64)** Length: 75–90 min/session Frequency: four sessions/12 weeks Duration: 12 weeks Supervised: yes **Relaxation (n = 59)** Length: 75–90 min/session Frequency: four sessions/week Duration: 12 weeks Supervised: yes **Control (n = 79)** Usual care	**Y:** 3.2 (2.4) **RX**: 3.7 (2.3) **C:** 3.5 (2.5) **Hedges’ g:** **Y:C** −0.12 (−0.45, 0.21) **RX:C** 0.08 (−0.26, 0.42) **Y:RX** −0.21 (−0.57, 0.14)
Chen 2013	Inter-treatment (RT)	China	0–III	Q: 45.3 ± 6.3 C: 44.7 ± 9.7	BFI	**Qigong (n = 49)** Length: 40 min/session Frequency: one sessions/week Duration: five to six weeks Supervised: yes **Control (n = 47)** Waiting list	**Q:** 3.1 (2.0) **C:** 2.7 (2.1) **Hedges’ g:** 0.19 (−0.21, 0.59)
Cornette 2016	Inter-treatment (CT)	USA	I–IIIB	(Median age) AE+RE: 52 C: 49	Multidimensional Fatigue Inventory (MFI)	**AE+RE (n = 20)** Length: 20–40 min/session Frequency: three sessions/week Duration: 27 weeks Supervised: no (home based) **Control (n = 22)** Usual care	**AE+RE:** 38 (12.3) **C:** 44.2 (13.9) **Hedges’ g:** −0.46 (−1.08, 0.15)
Courneya 2007	Inter-treatment (CT)	Canada	I–IIIA	Total Participants Range: 25–78 Mean: 49	Functional Assessment of Cancer Therapy—Anemia (FACT—An)	**AE (n = 68)** Length: 15–45 min/session Frequency: three sessions/week Duration: 18 weeks Supervised: yes **RE (n = 68)** Length: No information Frequency: three sessions/week Duration: 18 weeks Supervised: yes **Control (n = 60)** Usual care	**AE:** 42.1 (10.5) **RE:** 40.8 (10.5) **C:** 41.5 (9.8) **Hedges’ g:** **AE:C** 0.06 (−0.29, 0.41) **RE:C** −0.07 (−0.41, 0.27) **AE:RE** 0.12 (−0.21, 0.45)
Danhauer 2015	Inter-treatment (CT)	USA	Any stage	Y: 54.3 ± 9.6 C: 57.2 ± 10.2	Functional Assessment of Cancer Therapy—Fatigue (FACT—F)	**Yoga (n = 22)** Length: 75 min/session Frequency: one session/week Duration: 10 weeks Homework: 45 min/twice a week Supervised: yes **Control education (n = 18)** Length: 75 min/session Frequency: one sessions/week Duration: 10 weeks	**Y:** 39.8 (11.5) **C:** 32.6 (15.5) **Hedges’ g:** 0.51 (−0.26, 1.28)
Gokal 2016	Inter-treatment (CT)	UK	I–III	AE: 52.1 ± 11.7 C: 52.36 ± 8.9	FACT—F	**AE (n = 25)** Length: Began by completing 10 min of walking and then increased to 30 min/session Frequency: five sessions/week Duration: 12 weeks Supervised: no **Control (n = 25)** Usual care	**AE:** 26.04 (3.8) **C:** 33.6 (7.29) **Hedges’ g:** −1.28 (−1.89, −0.67)
Hu 2013	Inter-treatment (Immediate post-OP)	Taiwan	0–III	Total: 46.8 ± 9.7 AE: 46.5 ± 10.4 C: 47.1 ± 9.2	Functional Assessment of Chronic Illness Therapy—Fatigue (FACIT—F)	**AE (n = 30)** Length: 30–50 min/session Frequency: three to five sessions/week Duration: 5 weeks Supervised: no **Control (n = 25)** Usual care	**AE:** 38.2 (8.9) **C:** 36.9 (11.6) **Hedges’ g:** −0.12 (−0.62, 0.37)
Huang 2019	Inter-treatment (CT)	Taiwan	I–III	RE: 48.3 ± 7.9 C: 48.3 ± 8.7	BFI	**RE (n = 81)** Length: 30–40 min/session Frequency: five session/week Duration: 12 weeks Supervised: no **Control (n = 78)** Usual care	**RE:** 1.0 (2.2) **C:** 1.19 (2.3) **Hedges’ g:** −0.08 (−0.44, 0.27)
Husebø 2014	Inter-treatment (CT)	Norway	I–III	Total: 52.2 ± 9.3 AE+RE: 50.8 ± 9.7 C: 53.6 ± 8.8	Schwartz Cancer Fatigue Scale (SCFS-6)	**AE+RE** Length: 30 min/session Frequency: RE—three sessions/week AE—seven sessions/week Duration: 17 weeks Supervised: no **Control (n = 78)** Usual care	**AE+RE:** 12.01 (4.38) **C:** 13.13 (4.47) **Hedges’ g:** −0.25 (−0.76, 0.26)
Hwang 2008	Inter-treatment (RT)	Korea	NI	AE+RE: 46.3 ± 7.5 C: 46.3 ± 9.5	BFI	**AE+RE (n = 17)** Length: 50 min/session Frequency: three sessions/week Duration: five weeks Supervised: yes **Control (n = 20)** Usual care	**AE+RE:** 3.5 (1.7) **C:** 3.9 (2.1) **Hedges’ g:** −0.20 (−0.85, 0.45)
Jong 2018	Inter-treatment (CT)	Netherlands	I–III	Y: 51 ± 8.0 C: 51 ± 7.3	MFI	**Yoga (n = 39)** Length: 75 min/session Frequency: one session/week Duration: 12 weeks Homework: 5–20 min/daily Supervised: yes **Control (n = 29)** Usual care	**Y:** 14.6 (4.5) **C:** 14.2 (4.2) **Hedges’ g:** 0.09 (−0.39, 0.57)
Kirkham 2020	Inter-treatment (CT)	Canada	I–III	AE+RE: 51 ± 8.1 C: 49.5 ± 11	PFS	**AE+RE (n = 12)** Length: 25–40 min/session Frequency: three sessions/week Duration: 12 weeks Supervised: yes **Control (n = 15)** Waiting list	**AE+RE:** 4 (2.3) **C:** 3.9 (1.9) **Hedges’ g:** 0.05 (−0.72, 0.82)
Lee 2021	Inter-treatment (CT)	USA	I–III	Total: 46.9 ± 9.8	MFI−20	**RE (n = 15)** Length: 30 min/session Frequency: one session/week Duration: eight weeks Supervised: yes **Control (n = 15)** Usual care	**RE:** 54.3 (14.1) **C:** 49.3 (12.6) **Hedges’ g:** 0.36 (−0.36, 1.08)
Lötzke 2016	Inter-treatment (CT, HT, or RT)	Germany	I–III	Y: 51.0 ± 11.0 AE+RE: 51.4 ± 11.1	Cancer Fatigue Scale (CFS)	**Yoga (n = 45)** Length: 60 min/session Frequency: one to two sessions/week Duration: 12 weeks Supervised: yes **AE+RE (n = 47)** Length: 60 min per week Frequency: one session/week Duration: 12 weeks Supervised: yes	**Y:** 21.04 (9.91) **AE+RE:** 24.32 (10.63) **Hedges’ g:** −0.32 (−0.73, 0.10)
Mijwel 2018	Inter-treatment (CT)	Sweden	I–IIIA	RE: 54.4 ± 10.3 AE+RE: 52.7 ± 10.3 C: 52.6 ± 10.2	PFS	**RE (n = 70)** Length: 60 min/session Frequency: two sessions/week Duration: 16 weeks Supervised: yes **AE+RE (n = 74)** Length: 60 min/session Frequency: two sessions/week Duration: 16 weeks Supervised: yes **Control (n = 60)** Usual care	**RE:** 3.16 (2.92) **AE+RE:** 3.16 (2.61) **C:** 3.94 (2.95) **Hedges’ g:** **RE:C** −0.26 (−0.61, 0.08) **AE+RE:C** −0.28 (−0.63, 0.07) **RE:AE+RE** 0 (−0.37, 0.37)
Mock 2005	Inter-treatment (CT or RT)	USA	0–III	Total: 51.5 ± 9.3 AE: 51.3 ± 8.9 C: 51.6 ± 9.7	PFS	**AE (n = 60)** Length: 15–30 min/session Frequency: five to six sessions/week Duration: During treatments Supervised: no **Control (n = 59)** Usual care	**AE:** 3.5 (2.4) **C:** 3.7 (3.0) **Hedges’ g:** −0.07 (−0.45, 0.30)
Møller 2020	Inter-treatment (CT)	Denmark	I–III	Total: 51.7 ± 9.4 RE: 51.5 ± 9.6 C: 52.0 ± 9.3	EORTC QLQ-C30	**RE (n = 62)** Length: 20 min/session Frequency: two sessions/week Duration: 12 weeks Supervised: yes **Control Education (n = 59)** Health counseling and symptom guidance	**RE:** 58 (27) **C:** 59 (25) **Hedges’ g:** −0.04 (−0.39, 0.32)
Mostafaei 2021	Inter-treatment (CT)	Iran	0–III	RE: 48.5 ± 5.7 C: 49.6 ± 7.5	Fatigue Severity Scale (FSS)	**RE (n = 30)** Length: 20–30 min/session Frequency: three sessions/week Duration: six weeks Supervised: no **Control (n = 30)** Usual care	**RE:** 41.3 (9.4) **C:** 50.16 (9.96) **Hedges’ g:** −0.90 (−1.43, −0.37)
Mutrie 2007	Inter-treatment (CT or RT)	Scotland	0–III	Total: 51.6 ± 9.5 AE+RE: 51.3 ± 10.3 C: 51.8 ± 8.7	FACT—F	**AE+RE (n = 99)** Length: 45 min/session Frequency: 12 sessions/week Duration: During treatments Supervised: yes **Control (n = 102)** Usual care	**AE+RE:** 4 (10.4) **C:** 3.2 (12.1) **Hedges’ g:** 0.07 (−0.21, 0.35)
Naraphong 2015	Inter-treatment (CT)	USA	I–IIIA	AE: 46.4 ± 9.4 C: 47.2 ± 6.9	Piper Fatigue Scale—Revised (PFS—R)	**AE (n = 11)** Length: 30–40 min/session Frequency: three to five sessions/week Duration: 10 weeks Supervised: no **Control (n = 12)** Usual care	**AE:** 3.62 (2.07) **C:** 3.38 (2.75) **Hedges’ g:** 0.09 (−0.72, 0.91)
Schmidt 2015	Inter-treatment (CT)	Germany	I–IV	Total: 52.7 ± 10.0 AE+RE: 52.2 ± 9.9 RX: 53.3 ± 10.2	Fatigue Assessment Questionnaire (FAQ)	**AE+RE (n = 45)** Length: 60 min/session Frequency: two sessions/week Duration: 12 weeks Supervised: yes **Relaxation (n = 33)** Length: 60 min/session Frequency: two sessions/week Duration: 12 weeks	**AE:** 33.7 (18.8) **RX:** 41 (21.2) **Hedges’ g:** **AE+RE:RX** −0.36 (−0.81, 0.09)
Schmidt 2016	Inter-treatment (CT)	Germany	Early stage	RE: 53 ± 12.6 AE: 56 ± 10.2 C: 54 ± 11.2	MFI	**RE (n = 21)** Length: 60 min/session Frequency: two sessions/week Duration: 12 weeks Supervised: yes **AE (n = 20)** Length: 60 min/session Frequency: two sessions/week Duration: 12 weeks Supervised: yes **Control (n = 26)** Usual care	**RE:** 38.62 (17.43) **AE:** 48 (21.77) **C:** 43.52 (21.46) **Hedges’ g:** **RE:C** −0.24 (−0.82, 0.33) **AE:C** 0.20 (−0.38, 0.79) **RE:AE** −0.47 (−1.09, 0.15)
Steindorf 2014	Inter-treatment (RT)	Germany	0–III	Total: 55.8 ± 9.1 RE: 55.2 ± 9.5 RX: 56.4 ± 8.7	FAQ	**RE (n = 77)** Length: 60 min/session Frequency: two sessions/week Duration: 12 weeks Supervised: yes **Relaxation (n = 78)** Length: 60 min/session Frequency: two sessions/week Duration: 12 weeks	**RE:** 5.4 (2.3) **RX:** 5.9 (1.9) **Hedges’ g:** −0.24 (−0.55, 0.08)
Taso 2014	Inter-treatment (CT)	Taiwan	I–III	Total: 49.3 ± 10.2	BFI	**Yoga (n = 30)** Length: 60 min/session Frequency: two sessions/week Duration: eight weeks Supervised: yes **Control (n = 30)** Usual care	**Y:** 10.9 (6.9) **C:** 20.4 (5) **Hedges’ g:** −1.56 (−2.14, −0.97)
van Waart 2015	Inter-treatment (CT)	Netherlands	I–III	Total: 50.7 ± 9.1 AE+RE: 49.9 ± 8.4 RE: 50.5 ± 10.1 C: 51.6 ± 8.8	MFI	**AE+RE (n = 76)** Length: 50 min/session Frequency: two sessions/week Duration: 20 weeks Supervised: yes **AE (n = 77)** Length: 30 min/session Frequency: five sessions/week Duration: 20 weeks Supervised: no **Control (n = 77)** Usual care	**AE+RE:** 13.3 (4.7) **AE:** 11.7 (4.2) **C:** 14.7 (4.4) **Hedges’ g:** **AE+RE:C:** −0.31 (−0.62, 0.01) **AE:C** −0.69 (−1.02, −0.37) **AE+RE:AE** 0.36 (0.04, 0.68)
VanderWalde 2020	Inter-treatment (RT)	USA	0–III	(Median [range]) RE: 69 [66,67,68,69,70,71,72,73,74,75,76,77,78,79,80,81,82,83,84] AE: 68 [65,66,67,68,69,70,71,72,73,74,75,76,77,78,79,80,81,82,83]	FSI	**RE (n = 25)** Length: 30 min/session Frequency: four session/week Duration: six weeks Supervised: no **AE (n = 25)** Usual care	**RE:** 0 (8.9) **AE:** 0 (−1.7) **Hedges’ g:** −0.14 (−0.69, 0.42)
Vadiraja 2009	Inter-treatment (RT)	India	II–III	NA	EORTC QoL C30	**Yoga (n = 42)** Length: 60 min/session Frequency: three sessions/week Duration: six weeks Supervised: yes **Control education (n = 33)** Length: 15 min/session Frequency: one sessions/10 days Duration: six weeks	**Y:** 31.37 (21.79) **C:** 52.09 (24.24) **Hedges’ g:** −0.90 (−1.37, −0.42)
Wang 2011	Inter-treatment (CT)	USA	I–II	Total: 50.4 ± 9.6 AE: 48.4 ± 10.2 C: 52.3 ±8.8	FACIT—F	**AE (n = 35)** Length: 40 min/session Frequency: four sessions/week Duration: six weeks Supervised: no **Control (n = 37)** Usual care	**AE:** 45.81 (4.29) **C:** 39.91 (5.38) **Hedges’ g:** 1.19 (0.64, 1.74)
Wang 2014	Inter-treatment (CT)	China	NA	NA	CFS	**Yoga (n = 40)** Length: 50 min/session Frequency: four sessions/week Duration: 12 weeks Supervised: no **Control (n = 42)** Usual care	**Y:** 10.22 (2.06) **C:** 12.79 (2.06) **Hedges’ g:** −1.24 (−1.71, −0.76)
Aydin 2021	Post-treatment	Turkey	All stage	Total: 45.0 ± 2.2	EORTC QLQ—C30	**AE+RE (n = 24)** Length: AE—50 min/session RE—60 min/session Frequency: AE—three sessions/week RE—two sessions/week Duration: 12 weeks Supervised: yes (AE) **Control (n = 24)** Usual care	**AE+RE:** 34.2 (18.2) **C:** 38.4 (22.9) **Hedges’ g:** −0.20 (−0.77, −0.37)
Baglia 2019	Post-treatment	USA	0–III	AE: 62.0 ± 7.0 C: 60.5 ± 7.0	FACIT—Fatigue (Changed mean and SD are calculated by 95% CI)	**AE (n = 48)** Length: 50 min/session Frequency: three sessions/week Duration: 12 months Supervised: yes **Control** Usual care	**AE:** 0.5 (7.23) **C:** 5.7 (6.72) **Hedges’ g:** −0.74 (−1.15, −0.32)
Banasik 2011	Post-treatment	USA	II–IV	Y: 63.3 ± 6.9 C: 62.4 ± 7.3	Fatigue Score (Likert 0–4)	**Yoga (n = 7)** Length: 90 min/session Frequency: two sessions/week Duration: eight weeks Supervised: yes **Control (n = 7)** Regular routine	**Y:** 1 (0.89) **C:** 1.57 (0.98) **Hedges’ g:** −0.57 (−1.65, 0.51)
Bower 2012	Post-treatment	USA	0–II	Y: 54.4 ± 5.7 C: 53.3 ± 4.9	FSI	**Yoga (n = 16)** Length: 90 min/session Frequency: two sessions/week Duration: 12 weeks Supervised: yes **Control education (n = 15)** Length: 120 min/session Frequency: one session/week Duration: 12 weeks	**Y:** 3.4 (1.8) **C:** 4.9 (1.3) **Hedges’ g:** −0.93 (−1.67, −0.18)
Carson 2009	Post-treatment	USA	IA–IIB	Total: 54.4 ± 7.5 Y: 53.9 ± 9.0 C: 54.9 ± 6.2	Fatigue Subscale of Daily Menopausal Symptoms (SD is calculated via the t-value)	**Yoga (n = 17)** Length: 120 min/session Frequency: one session/week Duration: eight weeks Supervised: yes **Control (n = 20)** Waiting list	**Y:** 2.87 (0.39) **C:** 4.34 (0.39) **Hedges’ g:** −3.69 (−4.79, −2.59)
Cohen 2021	Post-treatment	USA	I–III	Total: 57.3 ± 8.8 RX: 59.7 ± 7.0 AE: 58.6 ± 10.4 AE+RE: 53.6 ± 8.0	PFS	**AE (n = 14)** Length: 20 min/session Frequency: three sessions/week Duration: 16 weeks Supervised: yes **AE+RE (n = 13)** Length: 20 min/session Frequency: three sessions/week Duration: 16 weeks Supervised: yes **Relaxation (n = 13)** Length: 20 min/session Frequency: three sessions/week Duration: 16 weeks	**RX:** 3.54 (1.56) **AE:** 4.15 (1.62) **AE+RE:** 2.65 (1.2) **Hedges’ g:** **AE:RX** 0.37 (−0.39, 1.13) **AE+AR:RX** −0.62 (−1.40, 0.17) **AE+AR:AE** 1.01 (0.20, 1.82)
Cramer 2015	Post-treatment	Germany	I–III	Total: 49.2 ± 5.0 Y: 48.3 ± 4.8 C: 50.0 ± 6.7	FACIT—F	**Yoga (n = 19)** Length: 90 min/session Frequency: one session/week Duration: 12 weeks Supervised: yes **Control (n = 21)** Usual care	**Y:** 42.8 (11.1) **C:** 37 (8.7) **Hedges’ g:** 0.57 (−0.06, 1.21)
Demello 2018	Post-treatment	USA	0–III	Total: 55.6 ± 9.6	FACIT-F	**RE (n = 38)** Length: 30 min/session Frequency: one session/week Duration: 12 weeks Supervised: yes **Control (n = 21)** Usual care	**RE:** 43.83 (7.28) **C:** 41.22 (8.49) **Hedges’ g:** 0.33 (−0.13, 0.78)
Dieli-Conwright 2018	Post-treatment	USA	0–III	Total: 53.5 ± 10.4	BFI	**AE+RE (n = 50)** Length: 30–50 min/session Frequency: three sessions/week Duration: 16 weeks Supervised: yes **Control (n = 50)** Usual care	**AE+RE:** 2.9 (1.5) **C:** 7.7 (2.4) **Hedges’ g:** −2.38 (−2.90, −1.86)
Do 2015	Post-treatment	Korea	0–III	AE+RE: 47.1 ± 8.5 C: 48.3 ± 8.2	FSS	**AE+RE (n = 32)** Length: 80 min/session Frequency: five sessions/week Duration: four weeks Supervised: yes **Control (n = 30)** Waiting list	**AE:** 17.8 (9.6) **C:** 37.1 (15) **Hedges’ g:** −1.52 (−2.0, −0.95)
Ergun 2013	Post-treatment	Tukey	Early stage	AE+RE: 49.6 ± 8.3 AE: 55.1 ± 6.9 C: 50.3 ± 10.4	BFI	**AE+RE (n = 20)** Length: 75 min/session Frequency: three sessions/week Duration: 12 weeks Supervised: yes **AE (n = 18)** Length: 30 min/session Frequency: three sessions/week Duration: 12 weeks Supervised: no **Control (n = 20)** Usual care	**AE+RE:** 2.86 (2.02) **AE:** 3.02 (2.5) **C:** 3.3 (1.79) **Hedges’ g:** **AE+RE:C** −0.23 (−0.85, 0.40) **AE:C** −0.13 (−0.75, 0.49) **AE+RE:AE** −0.07 (−0.69, 0.55)
Gal 2021	Post-treatment	Netherlands	NA	AE+RE: 58.0 ± 9.8 C:58.3 ± 9.5	MFI-20	**AE+RE (n = 127)** Length: 30 min/session Frequency: two sessions/week Duration: 12 weeks Supervised: yes **Control (n = 130)** Usual care	**AE+RE:** 10.4 (4.7) **C:** 10.3 (4.6) **Hedges’ g:** 0.02 (−0.24, 0.28)
Hagstrom 2016	Post-treatment	Australia	I–IIIA	Total: 59.1 ± 8.8 RE: 51.2 ± 8.5 C: 52.7 ± 9.4	FACIT-F	**RE (n = 19)** Length: 60 min/session Frequency: three sessions/week Duration: 16 weeks Supervised: yes **Control (n = 20)** Usual care	**RE:** 45.7 (7.57) **C:** 39.79 (10.36) **Hedges’ g:** 0.64 (−0.01, 1.28)
Jang 2021	Post-treatment	Korea	I–III	AE+RE: 49.9 ± 7.9 C: 47.6 ± 7.0	Korean version of the Revised Piper Fatigue Scale (K-R-PFS)	**AE+RE (n = 24)** Length: 60 min/session Frequency: one session/week Duration: 12 weeks Supervised: yes **Control (n = 20)** Usual care	**AE+RE:** 4.52 (1.93) **C:** 4.23 (2.18) **Hedges’ g:** 0.14 (−0.48, 0.76)
Kiecolt-Glaser 2014	Post-treatment	USA	0–IIIA	Total: 51.6 ± 9.2 Y: 51.8 ± 9.8 C: 51.3 ± 8.7	Multidimensional Fatigue Symptom Inventory Short Form (MFSI-SF)	**Yoga (n = 96)** Length: 90 min/session Frequency: two sessions/week Duration: 12 weeks Supervised: yes **Control (n = 90)** Waiting list	**Y:** 6.3 (20) **C:** 12.7 (20) **Hedges’ g:** −0.32 (−0.61, 0.03)
Kim 2020	Post-treatment	Korea	I–III	RE: 49.9 ± 7.6 C: 48.5 ± 6.8	K-R-PFS	**RE (n = 23)** Length: not mentioned Frequency: one session/week Duration: 12 weeks Supervised: yes **Control (n = 20)** Usual care	**RE:** 3.89 (1.19) **C:** 4.88 (1.52) **Hedges’ g:** −0.71 (−1.30, −0.12)
Littman 2012	Post-treatment	USA	0–III	Y: 60.6 ± 7.1 C: 58.2 ± 8.8	FACIT-F	**Yoga (n = 30)** Length: 75 min/session Frequency: five sessions/week Duration: 26 weeks Supervised: yes **Control (n = 27)** Usual care	**Y:** 45 (5.3) **C:** 43.1 (10.3) **Hedges’ g:** 0.23 (−0.29, 0.75)
Loh 2014	Post-treatment	Malaysia	I–II	NI (18–65 years)	FACIT-F	**Qigong (n = 32)** Length: 90 min/session Frequency: one session/week Duration: eight weeks Homework: 30 min/twice a week Supervised: yes **AE (n = 31)** Length: 90 min/session Frequency: one session/week Duration: eight weeks Supervised: yes **Control (n = 32)** Usual care	**Q:** 42.06 (6.04) **AE:** 41.81 (7.03) **C:** 40.38 (9.08) **Hedges’ g:** **Q:C** 0.22 (−0.28, 0.71) **AE:C** 0.17 (−0.32, 0.67) **Q:AE** 0.04 (−0.46, 0.53)
Milne 2008	Post-treatment	Australia	I–II	Total: 55.1 ± 8.2 AE+RE: 55.2 ± 8.4 C: 55.1 ± 8.0	Schwartz Cancer Fatigue Scale (SCFS)	**AE+RE (n = 29)** Length: 30 min/session (Aerobic), no information about resistance Frequency: three sessions/week Duration: 12 weeks Supervised: yes **Control (n = 29)** Waiting list	**AE+RE:** 11.9 (3.2) **C:** 17.4 (4.7) **Hedges’ g:** −1.35 (−1.92, −1.77)
Moraes 2021	Post-treatment	Brazil	NA	Total: 55.1 ± 8.2 RE: 55.2 ± 8.4 C: 55.1 ± 8.0	PFS	**RE (n = 12)** Length: not mentioned Frequency: one session/week Duration: eight weeks Supervised: yes **Control (n = 13)** Waiting list	**RE:** 2.3 (1.4) **C:** 3 (2.4) **Hedges’ g:** −0.34 (−1.13, 0.45)
Name 2015	Post-treatment	Thailand	0–IIIb	Total: ≤ 60 (n = 13) > 60 (n = 17)	FSI	**Qigong (n = 15)** Length: 60 min/session Frequency: four sessions/week Duration: 12 weeks Supervised: yes **Control (n = 15)** Usual care	**Q:** 17.33 (16.45) **C:** 28.8 (26.97) **Hedges’ g:** −0.50 (−1.23, 0.23)
Naumann 2012	Post-treatment	USA	I–III	AE+RE: 49.0 ± 8.2 C: 51.8 ± 11.5	PFS (Changed score, SD is calculated by SE)	**AE+RE (n = 11)** Length: 50 min/session Frequency: three sessions/week Duration: eight weeks Supervised: yes **Control (n = 10)** Usual care	**AE+RE:** −0.69 (1.49) **C:** −1.43 (1.33) **Hedges’ g:** 0.50 (−0.37, 1.37)
Ochi 2021	Post-treatment	Japan	I–II	RE: 49.0 ± 8.2 C: 51.8 ± 11.5	Cancer Fatigue Scale	**RE (n = 24)** Length: 30 min/session Frequency: three sessions/week Duration: 12 weeks Supervised: yes **Control (n = 24)** Usual care	**RE:** 17.5 (9.84) **C:** 19.96 (10.29) **Hedges’ g:** −0.24 (−0.81, 0.33)
Pagola 2020	Post-treatment	Spain	NA	AE+RE: 47 ± 7 RE: 51 ± 6	PERFORM questionnaire	**AE+RE (n = 13)** Length: 35 min/session Frequency: one to four sessions/week Duration: 16 weeks Supervised: yes **RE (n = 10)** Length: 70 min/session Frequency: one to four sessions/week Duration: 16 weeks Supervised: yes	**AE+RE:** 42 (12) **RE:** 50 (9) **Hedges’ g:** 0.74 (−0.12, 1.60)
Paulo 2019	Post-treatment	Brazil	I–III	AE+RE: 63.2 ± 7.1 RX: 66.6 ± 9.6	EORCT-QLQ-C30	**AE+RE (n = 18)** Length: 45 min/session Frequency: two sessions/week Duration: 36 weeks Supervised: yes **RX (n = 18)** Usual care Invited to participate in stretching and relaxation exercises	**AE+RE:** 0.6 (2.7) **RX:** 22.9 (15.8) **Hedges’ g:** −1.92 (−2.73, −1.12)
Pinto 2003	Post-treatment	USA	0–II	NA	Profile of Mood States (POMS)	**AE+RE (n = 12)** Length: 50 min/session Frequency: three sessions/week Duration: 12 weeks Supervised: yes **Control (n = 6)** Usual care	**AE+RE:** 9 (6.4) **C:** 4 (1.8) **Hedges’ g:** 0.88 (−0.15, 1.91)
Pinto 2008	Post-treatment	USA	0–II	AE: 53.2 ± 9.1 C: 52.9 ± 10.4	Fatigue 10 cm linear analog scale (Changed score, SD is calculated by SE)	**AE (n = 43)** Length: 30 min Frequency: five sessions/week Duration: 12 weeks Supervised: yes **Control (n = 43)** Usual care	**AE:** −14.93 (24.72) **C:** 1.79 (23.48) **Hedges’ g:** −0.69 (−1.12, −0.25)
Rogers 2014	Post-treatment	USA	I–II	Total: 56.2 ± 7.7 AE+RE: 57.2 ± 5.5 C: 55.2 ± 9.1	FSI	**AE+RE (n = 20)** Length: 40 min/session (Aerobic); no information about resistance Frequency: four sessions/week (Aerobic); 2 sessions/week (Resistance) Duration: 12 weeks Supervised: yes **Control (n = 22)** Usual care	**AE+RE:** 52.7 (5.4) **C:** 51.6 (6.9) **Hedges’ g:** 0.17 (−0.43, 0.78)
Rogers 2017	Post-treatment	USA	I–IIIA	Total: 54.4 ± 8.5	FSI	**AE (n = 110)** Length: 50 min/session Frequency: three sessions/week Duration: 12 weeks Supervised: yes **Control (n = 112)** Usual care	**AE:** 4 (1.8) **C:** 4.7 (2) **Hedges’ g:** −0.37 (−0.63, −0.10)
Saarto 2012	Post-treatment	Finland	NA	AE: 52.3 (range 32–68) C: 52.4 (range 35–68)	FACIT-F (Changed score, SD is calculated by 95% CI)	**AE (n = 263)** Length: 60 min/session Frequency: one session/week Duration: 52 weeks Supervised: yes **Control (n = 237)** Usual care	**AE:** 2.4 (8.69) **C:** 2.4 (8.64) **Hedges’ g:** 0 (−0.18, 0.18)
Santagenello 2020	Post-treatment	Brazil	I–III	RE: 52.1 ± 10.1 C: 59.0 ± 9.2	BFI	**RE (n = 11)** Length: 40 min/session Frequency: three sessions/week Duration: 12 weeks Supervised: yes **Control (n = 9)** Usual care	**RE:** 2.8 (2.1) **C:** 5.6 (2.5) **Hedges’ g:** −1.17 (−2.14, −0.20)
Schmidt 2017	Post-treatment	Germany	NA	AE+RE: 61.7 ± 10.0 C: 53.0 ± 10.7	FACIT-F	**AE+RE (n = 21)** Length: 60 min/session Frequency: two sessions/week Duration: 12 weeks Supervised: yes **Control (n = 28)** Usual care	**AE+RE:** 11.29 (4.15) **C:** 9.89 (3.82) **Hedges’ g:** 0.35 (−0.22, 0.92)
Stan 2016	Post-treatment	USA	0–II	Total: 62.1 ± 8.1 Y: 61.4 ± 7.0 RE: 63.0 ± 9.3	MFSI-SF (Changed score)	**Yoga (n = 18)** Length: 88 min/session Frequency: 3(−5) sessions/week Duration: 12 weeks Supervised: no **RE (n = 16)** Length: 26 min/session Frequency: three to five sessions/week Duration: 12 weeks Supervised: no	**Y:** −12.3 (14.5) **RE:** −7.4 (11.1) **Hedges’ g:** −0.37 (−1.05, 0.31)
Taylor 2018	Post-treatment	USA	Early stage	Y: 54.9 ± 8.8 C: 52.6 ± 8.2	BFI	**Yoga (n = 9)** Length: 75 min/session Frequency: one session/week Duration: eight weeks Supervised: yes **Control (n = 11)** Waiting list	**Y:** 1.85 (1.61) **C:** 2.1 (2.68) **Hedges’ g:** −0.11 (−0.99, 0.78)
Winters-Stone 2012	Post-treatment	USA	0–IIIA	Y: 68.6 ± 6.2 RX: 68.9 ± 2.9	Schwartz Cancer Fatigue (SCF)	**RE (n = 36)** Length: 60 min/session Frequency: three sessions/week (two 1 hr sessions were supervised; 1 hr session was home-based) Duration: 52 weeks Supervised: yes **Relaxation (n = 31)** Length: 60 min/session Frequency: three sessions/week (two 1 hr sessions were supervised and 1 hr session was home-based) Duration: 52 weeks	**RE:** 10.1 (4.7) **RX:** 9 (3.21) **Hedges’ g:** 0.27 (−0.22, 0.75)
Yagi 2015	Post-treatment	Turkey	I–II	RE: 62.3 ± 6.7 RX: 62.2 ± 2.9	Fatigue Visual Analog Scale (VAS)	**Yoga (n = 10)** Length: 60 min/session Frequency: one session/week Duration: eight weeks Supervised: yes **RE (n = 10)** Length: 60 min/session Frequency: one session/week Duration: eight weeks	**Y:** 2.86 (1.31) **RE:** 4.28 (0.97) **Hedges’ g:** −1.18 (−2.15, −0.21)
Yaʇli 2015	Post-treatment	Turkey	I–II	AE: 47.4 ± 7.6 Y+AE: 49.9 ± 4.7	FSS	**AE (n = 21)** Length: 30 min/session Frequency: three sessions/week Duration: six weeks Supervised: yes **AE + Yoga (n = 19)** Length: 30 min/session (Aerobic); 30 min/session Frequency: three sessions/week Duration: six weeks Supervised: yes	**AE:** 40.14 (7.58) **Y+AE:** 35.74 (5.99) **Hedges’ g:** 0.63 (−0.01, 1.26)
Yuen 2007	Post-treatment	USA	NA	AE: 53.1 ± 13.5 RE: 53.7 ± 11.3 C: 55.0 ± 13.4	PFS	**AE (n = 8)** Length: 20–40 min/session Frequency: three sessions/week Duration: 12 weeks Supervised: yes **RE (n = 7)** Length: No information Frequency: three sessions/week Duration: 12 weeks Supervised: yes **Control (n = 7)** Usual care	**AE:** 3.9 (1.71) **RE:** 2.79 (1.85) **C:** 4.16 (1.67) **Hedges’ g:** **AE:C** −0.14 (−1.16, 0.87) **RE:C** −0.73 (−1.82, 0.37) **AE:RE** 0.59 (−0.46, 1.63)

Abbreviations—OP: operation; CT: chemotherapy; RT: radiotherapy; HT: hormone therapy; C: control; RX: relaxation; RE: resistance; AE: aerobic exercise; AE+RE: aerobic exercise plus resistance; AY: aerobic yoga; Q: qigong; and Y: yoga.

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
