# Peer review of "Beneficial Exercises for Cancer-Related Fatigue among Women with Breast Cancer: A Systematic Review and Network Meta-Analysis"

_cancers, 2022, doi:10.3390/cancers15010151_

Round 1
Reviewer 1 Report
Thank you for your contribution, which is relevant to add knowledge to the current literature. Here my suggestion:
INTRODUCTION
- At the beginning, you introduced the role of CFR. I suggest to promote this paragraph with an overall overview about the impact on emotion and patients' abilities in the emotional field, which have a strong connection with the individual perception of fatigue (see: Durosini et al., 2022; Pourfallahi et al., 2020).
- Literature show the relevance of social connection in the involvement of physical activities and sport, also in the breast cancer field. Please, add details about it and, in general, explore better the benefits of physical activities (see: Durosini et al., 2021; Northey et al., 2019; Neil-Sztramko et al., 2019).
- As known, body strongly change during and after oncological treatments. I suggest to add more detials about this process on a psychological level (see: Sebri et al., 2022; Boquiren et al., 2013).
References:
- Boquiren, V.M.; Esplen, M.J.; Wong, J.; Toner, B.; Warner, E. Exploring the influence of gender-role socialization and objectified body consciousness on body image disturbance in breast cancer survivors.
- Durosini, I., Triberti, S., Sebri, V., Giudice, A. V., Guiddi, P., & Pravettoni, G. (2021). Psychological Benefits of a Sport-Based Program for Female Cancer Survivors: The Role of Social Connections. Frontiers in psychology, 12.
- Durosini, I., Triberti, S., Savioni, L., Sebri, V., & Pravettoni, G. (2022). The Role of Emotion-Related Abilities in the Quality of Life of Breast Cancer Survivors: A Systematic Review. International Journal of Environmental Research and Public Health, 19(19), 12704.
- Neil-Sztramko, S. E., Winters-Stone, K. M., Bland, K. A., & Campbell, K. L. (2019). Updated systematic review of exercise studies in breast cancer survivors: attention to the principles of exercise training. British journal of sports medicine, 53(8), 504-512.
- Northey, J. M., Pumpa, K. L., Quinlan, C., Ikin, A., Toohey, K., Smee, D. J., & Rattray, B. (2019). Cognition in breast cancer survivors: a pilot study of interval and continuous exercise. Journal of science and medicine in sport, 22(5), 580-585.
- Pourfallahi, M., Gholami, M., Tarrahi, M. J., Toulabi, T., & Kordestani Moghadam, P. (2020). The effect of informational-emotional support program on illness perceptions and emotional coping of cancer patients undergoing chemotherapy. Supportive Care in Cancer, 28(2), 485-495.
- Sebri, V., Durosini, I., Mazzoni, D., & Pravettoni, G. (2022). The Body after Cancer: A Qualitative Study on Breast Cancer Survivors’ Body Representation. International Journal of Environmental Research and Public Health, 19(19), 12515.
METHODS
- Did you involved or not grey literature, at the end? Why?
- Assessing your database, you did not involved pubmed or psycINfo. Why? Explain your choice, please
- Exclusion criteria are not completed. Please, add details about them. What about people who are not able to sign informed consent, for example? Other criteria?
- Please, add the letters refer to researchers who have conduucted each phases of "data extraction and article selection". The same for "risk of bias assessment", please.
- Just a clarification: when you stated "Post-treatment period", did you refer to survivorship? It is relevant to better focus your research study
- Please, delete the "risk of bias assessment" in Table 1 and make another Table for it. See other review to know how this Table must be done. In Table 1, add relevant outcomes of each study involved, please
DISCUSSION AND CONCLUSIONS
- I argue that it is needed to discuss on patients'motivation to participant in intervention. Motivation is a relevant construct that could impact on emotions and CRF too. I suggest to introduce it to exaplain results (see: Savioni et al., 2021; Sebri et al., 2022; Pesce et al., 2013).
References:
- Pesce, C., Crova, C., Marchetti, R., Struzzolino, I., Masci, I., Vannozzi, G., & Forte, R. (2013). Searching for cognitively optimal challenge point in physical activity for children with typical and atypical motor development. Mental Health and Physical Activity, 6(3), 172-180.
- Savioni, L., Triberti, S., Durosini, I., Sebri, V., & Pravettoni, G. (2022). Cancer patients’ participation and commitment to psychological interventions: a scoping review. Psychology & Health, 37(8), 1022-1055.
- Sebri, V., Durosini, I., Mazzoni, D., & Pravettoni, G. (2022). Breast Cancer Survivors’ Motivation to Participate in a Tailored Physical and Psychological Intervention: A Qualitative Thematic Analysis. Behavioral Sciences, 12(8), 271.
Author Response
Response to Reviewer 1
INTRODUCTION
- At the beginning, you introduced the role of CFR. I suggest to promote this paragraph with an overall overview about the impact on emotion and patients' abilities in the emotional field, which have a strong connection with the individual perception of fatigue (see: Durosini et al., 2022; Pourfallahi et al., 2020).
Response: Thank you for your comments. We simply add the above-mentioned reference to support our statement. Please refer to line 44
- Literature show the relevance of social connection in the involvement of physical activities and sport, also in the breast cancer field. Please, add details about it and, in general, explore better the benefits of physical activities (see: Durosini et al., 2021; Northey et al., 2019; Neil-Sztramko et al., 2019).
Response: Thank you for your comments. We simply add, “Also, PA will play an important role in social connections and help, peer influence group experience, and preserving positive experiential qualities as external circumstances” and the above-mentioned reference to support our statement. Please refer to the lines. 50-52
- As known, body strongly change during and after oncological treatments. I suggest to add more detials about this process on a psychological level (see: Sebri et al., 2022; Boquiren et al., 2013).
Response: Thank you for your comments. We agree with you, and it is an important analysis of the psychological impact of this review exercise because our aim is beneficial for physical fitness. Next time we hope to conduct the psychological impact of Yoga, and we will add your suggested reference.
References:
- Boquiren, V.M.; Esplen, M.J.; Wong, J.; Toner, B.; Warner, E. Exploring the influence of gender-role socialization and objectified body consciousness on body image disturbance in breast cancer survivors.
- Durosini, I., Triberti, S., Sebri, V., Giudice, A. V., Guiddi, P., & Pravettoni, G. (2021). Psychological Benefits of a Sport-Based Program for Female Cancer Survivors: The Role of Social Connections. Frontiers in psychology, 12.
- Durosini, I., Triberti, S., Savioni, L., Sebri, V., & Pravettoni, G. (2022). The Role of Emotion-Related Abilities in the Quality of Life of Breast Cancer Survivors: A Systematic Review. International Journal of Environmental Research and Public Health, 19(19), 12704.
- Neil-Sztramko, S. E., Winters-Stone, K. M., Bland, K. A., & Campbell, K. L. (2019). Updated systematic review of exercise studies in breast cancer survivors: attention to the principles of exercise training. British journal of sports medicine, 53(8), 504-512.
- Northey, J. M., Pumpa, K. L., Quinlan, C., Ikin, A., Toohey, K., Smee, D. J., & Rattray, B. (2019). Cognition in breast cancer survivors: a pilot study of interval and continuous exercise. Journal of science and medicine in sport, 22(5), 580-585.
- Pourfallahi, M., Gholami, M., Tarrahi, M. J., Toulabi, T., & Kordestani Moghadam, P. (2020). The effect of informational-emotional support program on illness perceptions and emotional coping of cancer patients undergoing chemotherapy. Supportive Care in Cancer, 28(2), 485-495.
- Sebri, V., Durosini, I., Mazzoni, D., & Pravettoni, G. (2022). The Body after Cancer: A Qualitative Study on Breast Cancer Survivors’ Body Representation. International Journal of Environmental Research and Public Health, 19(19), 12515.
METHODS
- Did you involved or not grey literature, at the end? Why?
Response: Thank you for your comments. We have searched grey literature for transparency for our search. It has been mentioned in the method section. However, we did not find a similar article. Please refer to lines 112-113
- Assessing your database, you did not involved pubmed or psycINfo. Why? Explain your choice, please
Response: Thank you for your comments. The national cheng Kung University database has covered PubMed or PsycINFO by Embase and Medline.
- Exclusion criteria are not completed. Please, add details about them. What about people who are not able to sign informed consent, for example? Other criteria?
Response: Thank you for your comments. This systematic review has used a published article, and we did not recruit participants and each individual article they mentioned about details of informed consent or others.
- Please, add the letters refer to researchers who have conduucted each phases of "data extraction and article selection". The same for "risk of bias assessment", please.
Response: Thank you for your comments. We simply add the researches name as a letter in the "data extraction and article selection" and "risk of bias assessment". Please refer line 129, 140,141 and 168
- Just a clarification: when you stated "Post-treatment period", did you refer to survivorship? It is relevant to better focus your research study
Response: Thank you for your comments. Yes, we consider survivorship. Please refer line 148
- Please, delete the "risk of bias assessment" in Table 1 and make another Table for it. See other review to know how this Table must be done. In Table 1, add relevant outcomes of each study involved, please
Response: Thank you for your comments. we change table 1. Added outcome variables and removed the risk of bias assessments. Please refer to table 1
DISCUSSION AND CONCLUSIONS
- I argue that it is needed to discuss on patients'motivation to participant in intervention. Motivation is a relevant construct that could impact on emotions and CRF too. I suggest to introduce it to exaplain results (see: Savioni et al., 2021; Sebri et al., 2022; Pesce et al., 2013).
Response: Thank you for your comments. We agree that assessing the qualitative perspective of patient motivation to participate in the exercise program is important. However, for this study, we did not focus on patients’ motivation to participate in the exercise. Next time we hope to conduct the patients’ motivation. We add in the future research section as well. We add your suggested reference Please refer line 501-502
References:
- Pesce, C., Crova, C., Marchetti, R., Struzzolino, I., Masci, I., Vannozzi, G., & Forte, R. (2013). Searching for cognitively optimal challenge point in physical activity for children with typical and atypical motor development. Mental Health and Physical Activity, 6(3), 172-180.
- Savioni, L., Triberti, S., Durosini, I., Sebri, V., & Pravettoni, G. (2022). Cancer patients’ participation and commitment to psychological interventions: a scoping review. Psychology & Health, 37(8), 1022-1055.
- Sebri, V., Durosini, I., Mazzoni, D., & Pravettoni, G. (2022). Breast Cancer Survivors’ Motivation to Participate in a Tailored Physical and Psychological Intervention: A Qualitative Thematic Analysis. Behavioral Sciences, 12(8), 271.
Reviewer 2 Report
Thank you for invitation for review this paper. In my opinion methods of this study is correct. But I suggest incorporate additional information about phase of cancer treatment ( when the patients were observe in presented studies). In my opinion it is very important information in aspect of CRF.
Many thank for well written limitations of this study.
Author Response
Response to Reviewer 2
Thank you for invitation for review this paper. In my opinion methods of this study is correct. But I suggest incorporate additional information about phase of cancer treatment ( when the patients were observe in presented studies). In my opinion it is very important information in aspect of CRF.
Many thank for well written limitations of this study.
Response: Thank you for your comments and very positive response
Round 2
Reviewer 1 Report
No other comments to add
Author Response
thank you
Reviewer 2 Report
Accept in current form.
Author Response
Response to Reviewer 2
"The reviewer 2 suggest incorporate additional information about phase of cancer treatment ( when the patients were observe in presented studies). "
Response: Thank you for your comments, and we added clarification in the methods section about the cancer treatment phase. Please refer to lines 143-145, 147-149, 152-153